# A Rare Case of Collision Tumours of the Ovary: An Ovarian Serous Cystadenoma Coexisting with Fibrothecoma

**DOI:** 10.3390/diagnostics12112840

**Published:** 2022-11-17

**Authors:** Michele Mongelli, Erica Silvestris, Vera Loizzi, Gennaro Cormio, Gerardo Cazzato, Francesca Arezzo

**Affiliations:** 1IRCCS Istituto Tumori “Giovanni Paolo II”, Department of Interdisciplinar Medicine (DIM), University of Bari “Aldo Moro”, Via Orazio Flacco 65, 70124 Bari, Italy; 2Section of Molecular Pathology, Department of Emergency and Organ Transplantation (DETO), University of Bari “Aldo Moro”, Piazza Giulio Cesare 11, 70124 Bari, Italy

**Keywords:** ovarian cancer, ultrasound, fibrothecoma, serous cystadenoma, collision tumour

## Abstract

The incidence of epithelial tumours of the ovary ranges from 9–17 per 100,000 and is the highest in high-income countries, with the exception of the Japan. The coexistence of neoplastic Müllerian epithelial and sex cord-stromal elements within a single tumour is extremely rare. We describe the case of a 74-year-old woman with a voluminous left adnexal formation. Pre-operative assessment with ultrasound evaluation made a suspicious diagnosis of benignity of the lesion. Bilateral salpingo-ovariectomy was performed. Intraoperative frozen section analysis results in the diagnosis of fibrothecoma in the context of serous cystadenoma. The diagnosis is confirmed by histological examination. Some authors suggest labelling this phenomenon as collision tumours.

**Figure 1 diagnostics-12-02840-f001:**
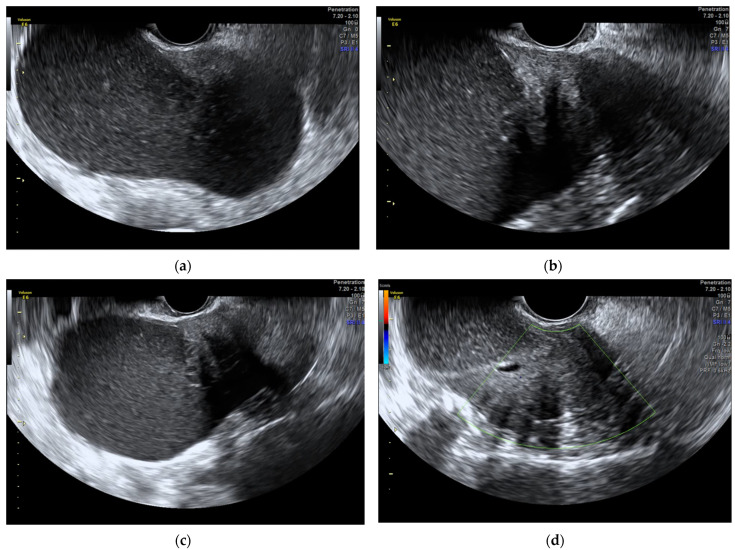
The ultrasound evaluation reveals a unilocular solid low-level cyst, measuring 128 × 65 mm. (**a**) The solid part, localized on the side of the cyst, has a smooth contour, regular internal echogenicity, and stripy shadows without cystic structures inside or on periphery of tumour; (**b**,**c**) CS 1 over the entire cyst (**d**). Upon probing, the cyst had a hard consistency. Ovarian tumours, as established by the WHO classification of the female genital tract, are classified in order of frequency into three groups: firstly, epithelial-stromal tumours; secondly, sex cord-stromal and steroid cell tumours; and, thirdly, germ cell tumours [1]. The coexistence of neoplastic Müllerian epithelial and sex cord-stromal elements within a single tumour is extremely rare. We report here a case of very unusual combination of fibrothecoma and serous cystadenoma in the left ovary of an elderly woman. A 74-year-old, multiparous, menopausal woman was admitted to our unit after the onset of sudden lymphoedema in the right leg. She had a CT scan in which a voluminous left adnexal formation measuring 14 × 9 cm with a solid token measuring 3.2 × 3.3 cm was found. The lesion compresses and dislocates the bladder. No other significant findings were noted. In the past, the patient underwent valve replacement with a biological valve for aortic valve insufficiency in 2017. During the last gynaecological visit in 2004, the objective examination and ultrasound scan did not reveal a pathological condition. Ultrasound evaluation was performed (Figure 1). Furthermore, CA-125, CA 19.9, CA 15.3, CEA, and AFP were found to be normal. The patient underwent surgery. Bilateral salpingo-ovariectomy was performed. Intraoperative frozen section analysis results in the diagnosis of fibrothecoma in the context of serous cystadenoma. (Figure 2). The diagnosis was confirmed by definitive histological examination (Figure 3). Pure thecoma is uncommon, accounting for no more than 1% of ovarian tumours. More frequently, thecoma presents as areas that resemble fibroma. In this case, it can be classified as fibrothecoma. Thecomas are typically unilateral and appear in postmenopausal women. No more than 10% arise in women younger than 30 years old [2].

**Figure 2 diagnostics-12-02840-f002:**
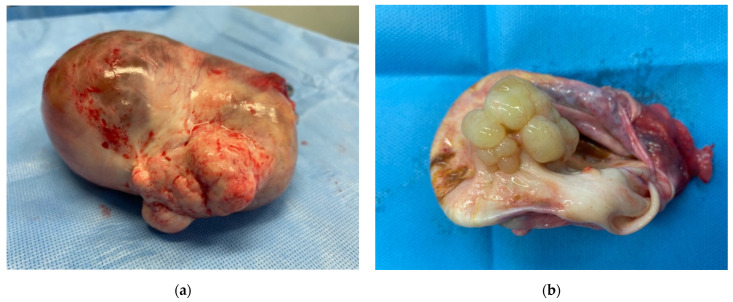
On pathologist evaluation, the cyst appears without surface proliferation (**a**). On cutting, abundant serous fluid leaks out and there are yellowish, translucent-looking mamelons inside (**b**). The serous tumour, instead, is the most common subtype of epithelial-stromal neoplasms. It tends to affect patients from the fourth to the sixth decade of their life [1,3].

**Figure 3 diagnostics-12-02840-f003:**
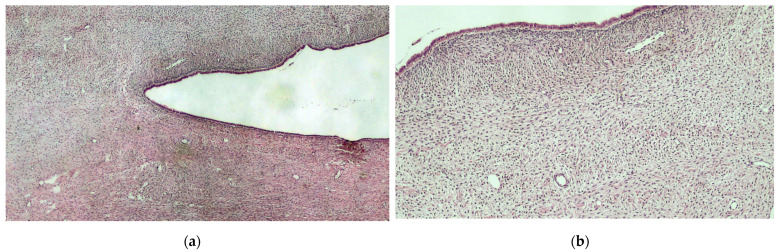
Histological preparation showing the presence of a monostratified epithelium (serous cyst) and uniform cell bundles, with pale-pink cytoplasm and poorly defined edges (thecoma). Hematoxylin-eosin, original magnification 4× (**a**). Histological preparation at higher magnification, showing the characteristic details of the lesion presented: fluid-filled sac coating subtended by stromal proliferation, consistent with ovarian thecoma. Hematoxylin-eosin, original magnification 10× (**b**). Furthermore, immunohistochemical staining revealed a p53 positivity in the monostratified epithelium of the serous cyst and α-inhibin positivity in the thecoma portion. Some authors suggest labelling this phenomenon as collision tumours. Collision tumours are defined as two contiguous but histologically different tumours, without mixture, in the same tissue or organ [4]. Collision tumours, in addition to the ovary, have been reported in several different organs, including oesophagus, stomach, thyroid, and brain, etc. [5,6,7,8]. Collision tumours of the ovary are quite rare. The most frequent histotypes combination are teratoma with mucinous tumour [4]. To date, only three cases of combined serous and thecal tumours have been reported in the literature. [9,10,11]. There are several hypotheses for the formation of collision tumours. According to the first hypothesis, the coexistence of two primary tumours in the same tissue is due to a ‘chance encounter’. As a second theory, it has been proposed that the presence of the first tumour changes the microenvironment, resulting in the development of the second primary tumour or the seeding of metastatic tumour cells. The third theory suggests that each primary tumour arises from a common stem cell [12]. Although there is no ultrasound definition to describe these mixed tumours, ultrasound can discriminate a benign formation from a malignant one, especially when performed by an experienced operator. In our case, the application of Simple Rules Risk Calculator (SRRISK) and the IOTA-adnex model, with 2.7% SSRisk estimated risk of malignancy and 91.2% patient-specific risk of benign tumour, respectively, allowed us to make a suspicious diagnosis of the benignity of the lesion [13,14]. The accuracy of the frozen section analysis is likewise very crucial, especially in young women who can be managed conservatively with fertility preservation. Collision tumours are extremely rare entities. As they are uncommon, a gynaecological and pathological evaluation in a gynaecological oncology referral centre is of crucial importance to ensure the most appropriate treatment possible.

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
