# Peer review of "A Rare Case of Collision Tumours of the Ovary: An Ovarian Serous Cystadenoma Coexisting with Fibrothecoma"

_diagnostics, 2022, doi:10.3390/diagnostics12112840_

Round 1

Reviewer 1 Report

The manuscript by Mongelli and co-authors describes a rare case of a collision tumor that has originated in the ovary. Conceptually, the manuscript could be of high importance to the medical professionals in the obstetrics/gynecology field of practice and research due to the rarity of this phenomenon.

Several critique points need to be addressed:

1. Immunostaining analysis with specific molecular markers associated with the serous histology of the epithelial tumor as well as those for thecoma is required.

2. The first sentence of the abstract needs clarification. Is ov.ca. the third most common by incidence or by mortality? also, are these statistics for one country, one region/continent, or worldwide?

3. Lines 35-36: the information regarding the last gyne onc visit is very vague and needs to be clarified.

4. The authors interchangeably use spelling "thecoma", "techoma", "fibrothecoma", and "fibrotechoma" throughout the manuscript. The correct terms need to be adhered to.

Author Response

Point 1:  Immunostaining analysis with specific molecular markers associated with the serous histology of the epithelial tumor as well as those for thecoma is required.

Response 1: we carried out immunostaining with the markers most commonly associated with serous histology and with thecoma respectively

Point 2: The first sentence of the abstract needs clarification. Is ov.ca. the third most common by incidence or by mortality? also, are these statistics for one country, one region/continent, or worldwide?

Response 2: we reported the worldwide incidence of epithelial ovarian cancer as reported in the latest figo cancer report of 2021

Point 3: Lines 35-36: the information regarding the last gyne onc visit is very vague and needs to be clarified.

Response 3: we reported a more detailed report of the last gynaecological examination performed during 2004 in another non-oncological centre

Point 4: The authors interchangeably use spelling "thecoma", "techoma", "fibrothecoma", and "fibrotechoma" throughout the manuscript. The correct terms need to be adhered to.

Response 4: We have changed and standardised the terminology as you suggested

Reviewer 2 Report

This is a well-written case report of some significance. I believe that it deserves to be published and am recommending acceptance. The authors should remember that a simple cyst is a fluid-filled sac, line 89. I recommend deleting "simple" at line 89. I would also like the results from the simple rules approach (SSISK)  and the IOTA ADNEX model either in the text or as a table.

Author Response

Point 1:  The authors should remember that a simple cyst is a fluid-filled sac, line 89. I recommend deleting "simple" at line 89. I would also like the results from the simple rules approach (SSISK)  and the IOTA ADNEX model either in the text or as a table

Response 1: Dear reviewer, we have modified line 89 and included the results of the SSRISK and IOTA ADNEX model either in the text as you suggested. 

Round 2

Reviewer 1 Report

My comments have been addressed.